# A general approach for stabilizing nanobodies for intracellular expression

John G Dingus, Jonathan CY Tang[†], Ryoji Amamoto, Grace K Wallick, Constance L Cepko*

Howard Hughes Medical Institute, Blavatnik Institute, Harvard Medical School, Boston, United States

**Abstract** Conventional antibodies and their derived fragments are difficult to deploy against intracellular targets in live cells, due to their bulk and structural complexity. Nanobodies provide an alternative modality, with well-documented examples of intracellular expression. Despite their promise as intracellular reagents, there has not been a systematic study of nanobody intracellular expression. Here, we examined intracellular expression of 75 nanobodies from the Protein Data Bank. Surprisingly, a majority of these nanobodies were unstable in cells, illustrated by aggregation and clearance. Using comparative analysis and framework mutagenesis, we developed a general approach that stabilized a great majority of nanobodies that were originally unstable intracellularly, without significantly compromising target binding. This approach led to the identification of distinct sequence features that impacted the intracellular stability of tested nanobodies. Mutationally stabilized nanobody expression was found to extend to in vivo contexts, in the murine retina and in *E. coli*. These data provide for improvements in nanobody engineering for intracellular applications, potentiating a growing field of intracellular interrogation and intervention.

*For correspondence: cepko@genetics.med.harvard. edu

Present address: †Columbia University, Columbia, United States

Competing interest: The authors declare that no competing interests exist.

## Editor's evaluation

This important study developed an innovative and powerful approach for improving the stability of nanobodies when expressed in an intracellular environment. The authors provide convincing evidence that by mutating key amino acids, they could enhance the stability of a majority of nanobodies in an intracellular environment without affecting their binding specificity. This study will be of general interest to a growing number of researchers using nanobodies as tools for their biological investigations.

## Introduction

Nanobodies, amino terminal fragments derived from a special class of antibody lacking light chains (*Hamers-Casterman et al., 1993*), are the smallest antibody derivatives that retain full antigen-binding function. Composed of a single variable domain of the heavy chain (V$_H$H), nanobodies boast several features that make them attractive tools for a range of applications. As monomers, they are versatile building blocks for protein engineering. Due to their compact binding interfaces, they have become invaluable as protein crystallization chaperones to resolve high-resolution crystal structures (*Muyldermans, 2013*; *Steyaert and Kobilka, 2011*; *Staus et al., 2014*). Their modular serum half-life and superior tissue penetration are attractive characteristics for the development of therapeutic biologics (*Van Audenhove and Gettemans, 2016*; *Jovčevska and Muyldermans, 2020*). Perhaps one feature that has been taken somewhat for granted is superior stability that facilitates intracellular expression. Full-length antibodies, as well as bulkier antibody fragments, are not normally amenable to intracellular expression, partially owing to the reducing environment of the cytoplasm that prevents

the formation of structurally crucial disulfide bonds. The ability to express nanobodies intracellularly opens the door for functional investigations of subcellular protein complexes and signaling pathways. Furthermore, intracellular nanobodies can be directed against novel therapeutic targets previously inaccessible to biologics.

Several groups have detailed intracellular nanobody expression to facilitate live imaging of subcellular factors (*Traenkle et al., 2015*; *Maier et al., 2015*; *Buchfellner et al., 2016*). However, few have commented on intracellular expression of nanobodies as a class as it pertains to stability. Those that have broadly detailed nanobody stability have done so in an extracellular context, in buffered solutions that do not resemble the cytoplasmic environment (*Kunz et al., 2017*; *Kunz et al., 2018*). We have been developing tools for nanobody-mediated, fluorescence-based sorting of live, target expressing cells (*Tang et al., 2016*), and genetic manipulation of cells expressing specific targets (*Tang et al.,*

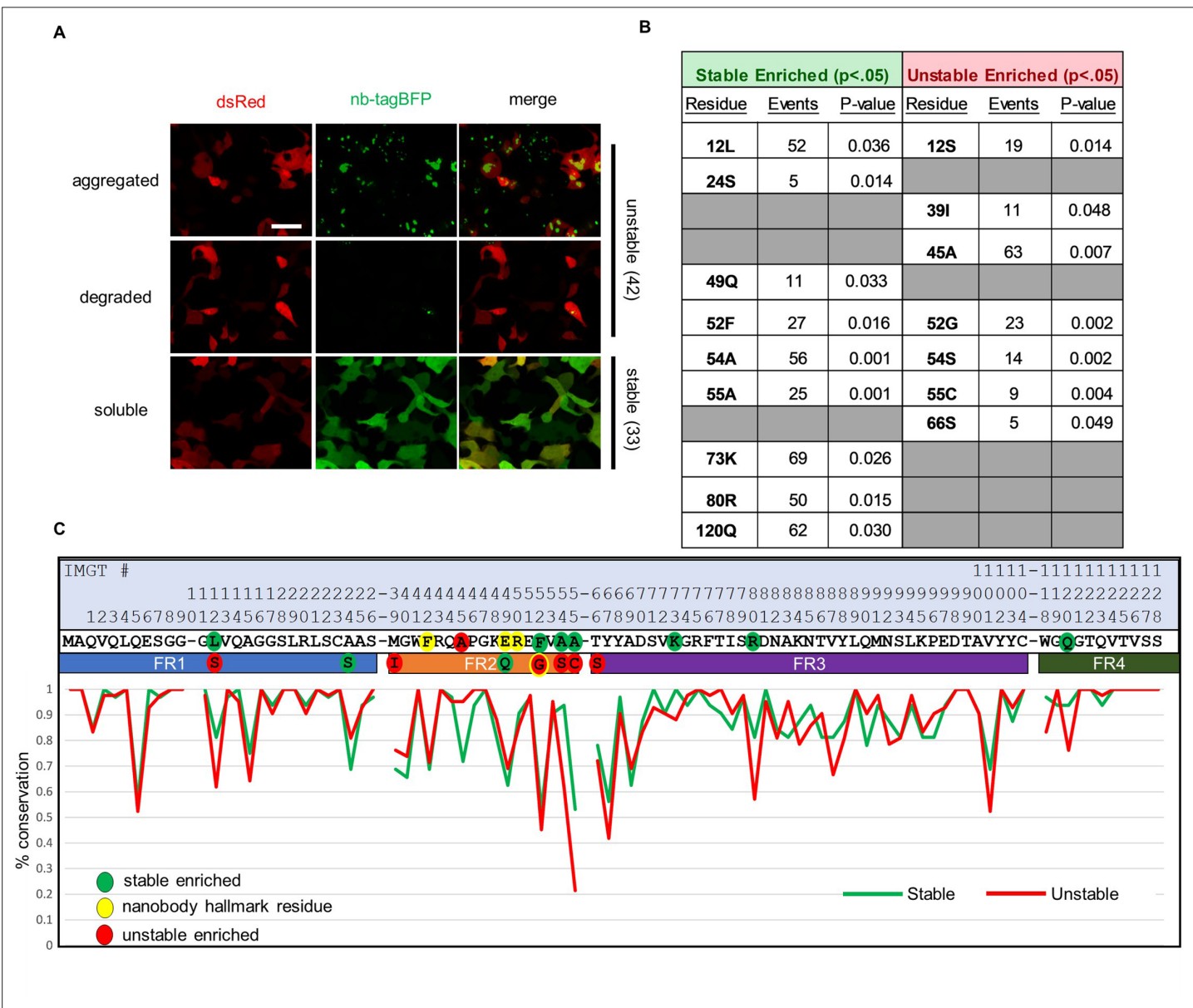

**Figure 1.** Sequence differences between intracellularly stable and unstable nanobodies. (**A**) Representative images of intracellularly stable and unstable nanobody-TagBFP fusions in transiently transfected 293T cells. Red signal is from co-transfected CAG-dsRed plasmid. (**B**) Significantly enriched positional residues between stable and unstable nanobodies (Fisher's exact test). Events denote the total number of instances of each positional residue across all nanobodies (75 total). Gray cells denote positional insignificance. (**C**) Positional amino acid conservation across framework sequences of stable and unstable nanobodies. Scale bar in Panel A is 20 µm.

*2013*; *Tang et al., 2015*). As we expanded our initial studies, we found that many nanobodies were not natively stable in the intracellular environment. We thus set out to investigate intracellular expression of nanobodies more systematically. To this end, 75 unique nanobody sequences from crystal structures uploaded to the Protein Data Bank (PDB) were cloned into mammalian expression vectors, fused C-terminally to a fluorescent protein (FP). Transfection and live-cell fluorescence imaging of these nanobody-FP fusions in both 293T and HeLa cells revealed that many nanobodies degrade and/or aggregate within the cytoplasm and nucleus, while others appear stable and soluble (*Figure 1A*). Following these observations, we set out to define intracellular instability based on sequence features, as well as to derive a standardized method for stabilization of previously intracellularly unstable nanobodies through framework mutagenesis. By leveraging positional sequence conservation apparent across intracellularly stable nanobodies, we have derived a method by which the vast majority of intracellularly unstable nanobodies can be stabilized for intracellular expression. This mutational stabilization, first observed in cell lines, was further tested for expression in vivo, in *E. coli*, and in the murine retina, whereupon it was found to be effective. These findings will contribute to the broader adoption of nanobodies as powerful reagents for both research and therapeutic applications.

## Results
### Classification of intracellularly stable and unstable nanobodies
To investigate nanobody intracellular expression broadly, we first compiled a repertoire of nanobody sequences to profile. The PDB was manually combed for nanobody crystal structures and sequences, resulting in a list of 75 unique sequences, representing nanobodies derived from 3 camelid species and targeting 44 unique protein targets (sequences and targets detailed in *Supplementary file 1*). These sequences were cloned into a mammalian expression vector in frame with TagBFP, linked by a 2 amino acid linker. All 75 sequences were expressed via transient transfection in both 293T and HeLa cells in separate wells of a 96-well plate. Intracellular expression patterns were captured by live fluorescence imaging. 33 out of 75 sequences exhibited strong, diffuse fluorescent signal, expected of well-expressed and intracellularly stable and soluble protein. Interestingly, 42 out of 75 sequences exhibited intracellular phenotypes suggestive of intracellular instability, including low or absent fluorescent signal coincident with varying degrees of aggregation (*Figure 1A*). These results were confirmed over several replicate experiments.

After binning the nanobodies into 'stable' and 'unstable' groups, their sequences were analyzed for distinguishing features. Nanobodies are composed of three variable loops (CDR1–3), responsible for the majority of target-specific interaction, and four framework regions (FR1–4), forming the conserved framework structure of the nanobody. First, the average length of variable loop CDR3 was calculated for each group, as CDR3 loop length is highly variable (between 5 and 26 amino acids for nanobodies in the examined repertoire) and CDR3 represents the area of both the greatest sequence-level and structural diversity across nanobodies. While there was little difference in average CDR3 loop length between the groups (16.4 amino acids for the stable group and 16.9 amino acids for the unstable group), one difference related to CDR3 stood out: 17/75 nanobodies contained a CDR3 cysteine that normally forms a disulfide bond tethering CDR3 to either CDR1 (first variable loop) or FR2 (second framework region); all 17 were in the unstable group (*Figure 2*). Because disulfides do not form when nanobodies are expressed in the cytoplasm, this feature may represent a structural liability for intracellular expression.

We next derived consensus sequences (the sequence constructed from the most common amino acid at each position across nanobody sequences) of the framework regions for both stable and unstable nanobodies. Sequence-level deviation between each stability-binned nanobody and the consensus sequence for stable nanobodies is illustrated in *Figure 2*. Stable and unstable consensus sequences for the framework region were identical except at one position, amino acid 52 (IMGT numbering), where the most common amino acid across stable nanobodies was Phe and across unstable nanobodies was Gly. Interestingly, position 52 is one of four hallmark framework positions cited as differentiating VHs (heavy chain variable regions of conventional antibodies) and V$_H$Hs (*Muyldermans, 2013*) (heavy chain variable regions of heavy-chain-only antibodies, analogous to nanobody). These four positions normally contribute to a hydrophobic interaction interface between VH and VL (light chain variable region) in conventional antibodies, and are substituted for amino acids

☐ extra disulfide

**Unstable NBs**

```
IMGT #
                                                                          11111-11111111111
             11111111222222222-34444444444555555-666677777777778888888888899999999999900000-11222222222
           123456789012345678901234566-90123456789012345-678901234567890123456789012345678901234-89012345678
           MAQVQLQESGG-GLVQAGGSLRLSCAAS-MGWFRQAPGKEREFVAA-TYYADSVKGRFTISRDNAKNTVYLQMNSLKPEDTAVYYC-WGQGTQVTVSS
```

| ID | sequence variability | animal | CDR3 length |
|---|---|---|---|
| 3ZKQ | - P - T V G W ST- T R L -R P | llama | 5 |
| 3K1K | V -A P - R Y W G-SS E D R - | dromedary | 8 |
| 3K7U | - T - -LF N R T - | llama | 10 |
| 3CFI | - P - S V GL W SG- S TAP IL R -R | llama | 11 |
| 4LGP | V T - P T G -I H WLVC- V V A L D GI - | alpaca | 11 |
| 4MQS | - D -I Q G SC- I A S E V - | llama | 11 |
| 1ZV5 | D V - S E -I D G VF- Q S - | dromedary | 14 |
| 2P42 | V - - G - L KG D T - | dromedary | 14 |
| 3V0A | V - P - S V EGF W SS-AWDG A T D L SN Q G - | llama | 14 |
| 4QKX | - - Y Q L - N N A - | llama | 14 |
| 4WEU | - - K Y - IA L D - | llama | 14 |
| 4X7C | D V - P - Y Q L S- N G -R R | alpaca | 14 |
| 1KXV | V - T P - S Y R G D SG- T V A QG A D D M - | dromedary | 15 |
| 4LHJ | V - - Y L T-SN G S - | alpaca | 15 |
| 4OCL | - P VD - A T- R - R | llama | 15 |
| 4WEM | - E - Q GY-LN G F SN S G F- K | llama | 15 |
| 4WEN | - P T - Y SK H- EF T D - K | llama | 15 |
| 5IV0 | V - P T -I G SC- FMN D I - | alpaca | 15 |
| 1KXQ | V - S S -V G - S L Q N GI - | dromedary | 16 |
| 4HEM | V - - P T- RN NM - | llama | 16 |
| 4LAJ | V - P -I G SC- -Y K | llama | 16 |
| 4C58 | - S G - A G C-S S R Q T AF L S I -A | llama | 17 |
| 4KML | - P - G S-SD T M N T - | llama | 17 |
| 4KRM | K E - S T T - SG- G D I - | llama | 17 |
| 2X6M | G V - S - R G R- A D E I - | dromedary | 18 |
| 4W6X | - S T - G C-S Q D F R I - | llama | 18 |
| 3K74 | - P - Y V R GL W SM- K E L TS -K | llama | 19 |
| 4GRWf | E V - P -I G SC- ES - | llama | 19 |
| 4W6W | - S - G C-VN Q S K L E L S - | llama | 19 |
| 4I13 | - A -I - V GE E I MN V R N - R | llama | 20 |
| 4S10 | - - S- R A K S DN N D - | llama | 20 |
| 1G6V | V - S - G T- G Q I -R | dromedary | 21 |
| 1KXT | VA - S - Y C LS R-AN A A D - | dromedary | 21 |
| 1RJC | E A - S Q T - G V- A Q L L L M - | dromedary | 21 |
| 3JBC | - S T - G G-A Q K I - | dromedary | 21 |
| 4I0C | - S E - A G -P Q RM E M - | dromedary | 21 |
| 4LGS | V - S - A S- L AL N - | alpaca | 22 |
| 4QGY | V - -I G SC- P A S K M - K | llama | 22 |
| 4W6Y | - S - A V G S- S T - | llama | 22 |
| 4HEP | D V - P E -I G SY- V T S L - K L | llama | 24 |
| 3G9A | D - S - A C L SN- T G D VN S R - K | dromedary | 25 |
| 1JTP | D A - S - G - Q L E I - | dromedary | 26 |
```
```

**Stable NBs**

```
IMGT #
                                                                          11111-11111111111
             11111111222222222-34444444444555555-666677777777778888888888899999999999900000-11222222222
           123456789012345678901234566-90123456789012345-678901234567890123456789012345678901234-89012345678
           MAQVQLQESGG-GLVQAGGSLRLSCAAS-MGWFRQAPGKEREFVAA-TYYADSVKGRFTISRDNAKNTVYLQMNSLKPEDTAVYYC-WGQGTQVTVSS
```

| ID | sequence variability | animal | CDR3 length |
|---|---|---|---|
| 2XT1 | V - A Y A LI - V D T DD IL D M - | alpaca | 6 |
| 4X7F | D V - P - A Y G EQ L V- D M L SN R - | alpaca | 7 |
| 4ORZ | - Y Q L F- D P V - S | llama | 10 |
| 4EIG | - K - T Y L L- MTV VQ E N - | llama | 11 |
| 4CDG | - - A Y T Q RI I- N V D M I - R | llama | 13 |
| 4LGR | V - P H -TC Y GT Q L - ID - | alpaca | 13 |
| 4M3K | - P - Y D G L L- T G E S - | llama | 14 |
| 3P0G | - - Y Q L - N N A - | llama | 15 |
| 4C57 | - P S - S V RV GL W G-AH R ML S SD GL -SS | llama | 15 |
| 4WGV | - -AN Y P MQ L T-AN R R G - | llama | 15 |
| 2BSE | - T - LA P L V - V SG I - | llama | 16 |
| 4GRWh | E V - -V -P D R L - | llama | 16 |
| 4IOS | V - D V -I - GR M CAA A L - | llama | 16 |
| 4NBX | V - A - A P - | llama | 16 |
| 4NC2 | V - - -PN S Q - | llama | 16 |
| 4P2C | - V - Y Q S- N R - | llama | 16 |
| 4QO1 | - - R L Q- E D T E N NAD GI F - | llama | 16 |
| 1OP9 | - S S -S G - Q M - E | dromedary | 17 |
| 3EBA | V - S S -S GL W - Q M - | dromedary | 17 |
| 4AQ1 | - - G- GA M G A - | llama | 17 |
| 4GFT | - T P K S A- Q G-Q M A V A - | llama | 17 |
| 4NBZ | K E - -VA A V- V N R F - | llama | 17 |
| 1ZVH | D V - S -L G -P V L E L - | dromedary | 18 |
| 4GRWe | E V - P -IA G SG-A S R - K L | llama | 18 |
| 1ZVY | D V - S - T A G - T Q K MA R D V S I T - | dromedary | 20 |
| 3RJQ | - T - - D A K A - | llama | 20 |
| 4DK3 | - - - T E - | llama | 20 |
| 4N1H | - A K - - K A D W R - | llama | 20 |
| 4TVS | V - -V L T- H - | alpaca | 20 |
| 4EIZ | - T - T - L M - | llama | 21 |
| 4FHB | - E - D V- A I - | llama | 22 |
| 4LHQ | V T - T T S - - R V HL L A - | alpaca | 22 |
| 4KRO | - P - Q - T T - | llama | 23 |
```

**Figure 2.** Framework sequence variability (compared to consensus sequence) across stable and unstable nanobodies. Note: All nanobodies contain a conserved set of cysteines that normally forms a disulfide bond through the hydrophobic core of the nanobody (23Cys and 104Cys). Highlighted in yellow, 'extra disulfide bond' refers to nanobodies with an additional pair of cysteines, one of which is always located in CDR3, that normally forms a disulfide bond that impacts CDR3 conformation.

*Figure 2 continued on next page*

*Figure 2 continued*

The online version of this article includes the following source data for figure 2:

**Source data 1.** Editable version of *Figure 2*.

that increase the hydrophilic character of the surface in V$_H$Hs. 52Gly has been shown to enhance nanobody solubility, but at the expense of protein yields in *E. coli*, which may point to decreased stability (*Davies and Riechmann, 1994*; *Davies and Riechmann, 1995*).

Next, we examined total positional enrichment for each amino acid across each position for both stable and unstable nanobody frameworks. Applying a Fisher's exact test, 12 framework positions with residues that were significantly enriched in either group were identified, with 4 of those positions having a specifically enriched residue in both groups (*Figure 1B*). Of note, 52Gly was strongly enriched in the unstable group (19/23 occurrences in unstable nanobodies), as well as 55Cys, a cysteine that normally forms a disulfide bond with a CDR3 cysteine in a subset of nanobodies, as noted above (9/9 occurrences in unstable nanobodies). Additionally, 54Ser emerged as a strongly enriched unstable residue likely to drive instability (13/14 occurrences in unstable nanobodies). Position 54 points its functional group into the hydrophobic core of the nanobody, and is usually occupied by a small hydrophobic amino acid (93% of stable nanobodies examined contain 54Ala). A hydrophilic substitution at position 54 is likely to contribute to destabilization of the hydrophobic core.

Each unstable nanobody had at least one of seven residues that were identified as significantly enriched in the unstable group. However, the most common positional residue enriched in the unstable group, 45Ala, is the most common residue at that position for both groups, and is therefore unlikely to be a strong driver of instability (40/42 unstable nanobodies and 24/33 stable nanobodies contain 45Ala). 10/42 unstable nanobodies had 45Ala as their only enriched residue. Since this preliminary sequence analysis was unlikely to comprehensively explain the differences between stability groups, we decided to apply a broad mutagenesis strategy to unstable nanobodies to try to achieve stability.

## Stabilizing mutagenesis based on positional conservation

We set out to define a general mutagenesis approach to stabilize most, if not all, nanobodies for intracellular expression, taking advantage of sequence features observable across expression-profiled nanobodies. At the outset, our strategy was based upon the assumption that sets of important stabilizing residues are likely to be highly conserved across intracellularly stable nanobodies. While the framework consensus sequences for stable and unstable nanobodies are nearly identical, positional conservation varies between the two groups (*Figure 1C*). Our approach for conservation-based mutagenesis is schematized in *Figure 3A*. A threshold of ≥80% positional conservation was applied to generate a partial consensus sequence of the most highly conserved positional residues across stable nanobodies. This partial consensus framework was applied as a filter to identify non-conforming positional residues in each unstable nanobody for mutagenesis. At each position that a given unstable nanobody disagreed with the filter, that residue was changed to agree with the filter. An exception was made to maintain any cysteine that normally participates in a disulfide bond, as these cysteines likely impact positioning of CDR3. Applying these changes to all 42 intracellularly unstable nanobodies yielded mutants with a range of between 2 and 12 changes per nanobody, with an average mutational load of 5.7 changes per nanobody. Mutation numbers for each nanobody are reported in *Figure 4*, *Figure 5* and *Figure 6*.

We cloned the derived mutant nanobodies and expressed them as FP fusions in 293T and HeLa cells as we had expressed their wild-type parents, previously. Sequence variability before and after mutagenesis for unstable nanobodies, as well as stability status following mutagenesis, is illustrated in *Figure 4*. Fluorescent imaging revealed that 26/42 nanobodies had been effectively stabilized via targeted mutagenesis, as illustrated by strong fluorescent signal and a total absence of punctate aggregates (*Figure 7A, C*). Another 6/42 nanobodies exhibited improvement compared to their wild-type parent with respect to aggregation levels and soluble protein fraction. 10/42 nanobodies exhibited no improvement from mutagenesis.

Aligning the 42 mutant sequences, grouping sequences that were effectively, partially, or not stabilized via mutagenesis, a striking correlation emerged: 13/16 nanobodies that were not effectively stabilized via mutagenesis had 52Gly (*Figure 5*). Because position 52 is a relatively variable framework position, it was not included in the conservation filter for first-pass mutagenesis. Additionally,

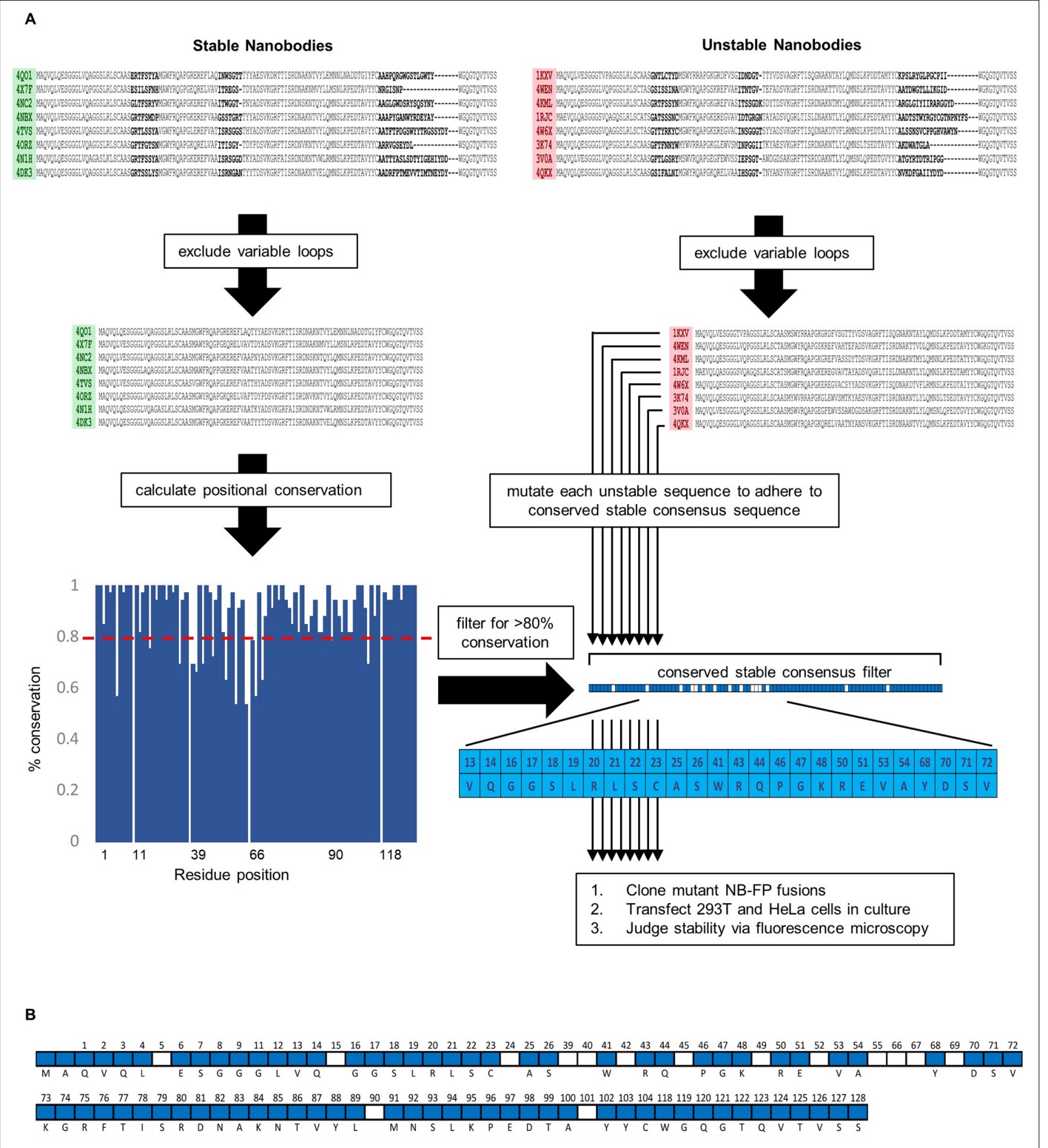

**Figure 3.** Schematic overview of conservation-based stabilizing mutagenesis strategy. (**A**) Nanobody-TagBFP fusions were classified as 'stable' or 'unstable' based on intracellular expression via transient transfection in 293T cells. Amino acid sequences were binned according to stability group, and variable domain sequences (CDRs) were excluded from downstream consideration. Positional sequence conservation was calculated across stable nanobody sequences, and positional amino acids of high conservation (>80%) were compiled to form a partial consensus filter. Each individual unstable nanobody framework was then compared to this filter, and any positional amino acid disagreement was resolved to adhere to the filter. Mutated

*Figure 3 continued on next page*

*Figure 3 continued*

nanobodies were then cloned, transfected into 293T and HeLa cells, and judged for stability via fluorescence microscopy. (**B**) All positional amino acids captured in the partial consensus framework (blue cells). White cells denote framework positions excluded from the partial consensus framework.

The online version of this article includes the following figure supplement(s) for figure 3:

**Figure supplement 1.** Nanobody mutagenesis flow chart and species consensus comparison.

non-consensus residues at positions 90 and 101 were enriched in non-stabilized nanobodies. For nanobodies not effectively stabilized by stage 1 mutagenesis, up to two additional changes were made wherever possible, Gly52Phe and X90Gln, to try to achieve stability. Because deviation at position 101 was almost perfectly correlated with the presence of Gly52, a likely driver of instability, this position was not changed (*Figure 4*). As a result of this additional round of mutagenesis, 11/13 additional nanobodies were effectively stabilized. In total, 37/42 previously intracellularly unstable nanobodies were rendered stable and soluble (*Figure 7C*).

## Identification of strong drivers of intracellular stability

Encouraged by the success of the stabilizing mutagenesis approach, we set out to identify specific drivers of stability from among the partial consensus framework. To do this, we sought to eliminate 'passenger' mutations for individual stabilized nanobodies and to identify the minimal, necessary mutational subsets required for intracellular stabilization. Five mutationally stabilized nanobodies with high mutational load and the high-confidence stability driver Ser54Ala were chosen for further examination (3K74, 3V0A, 3G9A, 4MQS, and 1KXV). 1KXT, a mutationally stabilized nanobody receiving the unique mutation Ser53Val, was also examined as both positions 53 and 54 point inward to the hydrophobic core of the nanobody. Subsets of the original mutations imposed to achieve stability were chosen, and new mutant variants were generated for intracellular stability testing. Mutation subsets were chosen based on crystal structure data and our own stabilization statistics. Several variants are depicted in *Figure 7D, E*. After testing several mutation variants for each nanobody, Ser54Ala was found to be sufficient to stabilize 3K74 on its own (originally stabilized with eight mutations). Ser53Val similarly stabilized 1KXT by itself. 3V0A and 4MQS required only Ser54Ala and one additional mutation each to achieve stability (originally stabilized with 11 and 7 mutations, respectively). The minimal necessary sets of mutations needed to stabilize 3G9A and 1KXV were not identified, suggesting that several of their original mutations, in addition to Ser54Ala, were necessary for stabilization.

68Tyr and 80Arg were identified in nanobodies 3V0A and 4MQS, respectively, as important for their stabilization. These residues were subsequently investigated for their role in the stabilization of three additional nanobodies not originally containing the strong instability driver 54Ser: 4WEN, 4LHJ, and 1KXQ. 4LHJ was originally stabilized from only two mutations (Gln80Arg and Ser88Tyr). However, removal of Ser88Tyr did not destabilize 4LHJ, illustrating the stabilizing influence of 80Arg alone (data not shown). Surprisingly, 4WEN was effectively stabilized by 68Tyr alone, when it had originally received five mutations (*Figure 7D, E*). 1KXQ, which originally received both Ser68Tyr and Gln80Arg mutations in addition to five others, was effectively stabilized by 68Tyr alone, again illustrating the importance of 68Tyr for stability and suggesting only circumstantial importance of 80Arg (data not shown).

## Mutationally stabilized nanobodies retain target-binding function in cells

It is crucial for stabilizing mutagenesis not only to facilitate intracellular expression, but also to maintain the nanobody's ability to bind its target. Conventional antibodies engage their targets in a largely stereotyped fashion, relying heavily on variable loops for target interaction, with minimal direct framework contribution. In contrast, nanobodies exhibit a greater paratope diversity, and contribution of framework residues to binding is more common (*Mitchell and Colwell, 2018*). We compiled crystal structure data, curated by EMBL-EBI and made available through PDBe PISA, describing the interaction interfaces between nanobodies and their targets (*Supplementary file 2*). Examination of these interfaces revealed that the great majority of interacting residues across nanobody framework regions are located in the most highly variable framework positions, positions omitted from our partial consensus mutagenesis approach (*Figure 8—figure supplement 1A*). While the possibility

| | unstable |
| | partially stable |

```
IMGT #                                                                    1111111111111111
                 1111111111222222223444444444455555556666777777777788888888889999999999000001122222222
                 123456789123456789012345690123456789012345678901234567890123456789012345678901234890123456780
NB#              MAQVQLQESGGGLVQAGGSLRLSCAASMGWFRQAPGKEREFVAATYYADSVKGRFTISRDNAKNTVYLQMNSLKPEDTAVYYCWGQGTQVTVSS
```

| NB# | ID | Sequence variants |
|-----|------|---------|
| 1 | 1G6V | V  S              G  T  G          Q              I  R |
| 2 | 1JTP | D  A  S           G              Q         L  E    I |
| 3 | 1KXQ | V  S    S   V     G    S      L Q  N            GI |
| 4 | 1KXT | VA  S           Y    C LS RAN        A  A   D |
| 5 | 1KXV | V  T P        S Y R    G D  SG T V   A     QG    A    D    D  M |
| 6 | 1RJC | E  A  S  Q     T       G  V A    Q L  L     L         M |
| 7 | 1ZV5 | D  V  S  E      I      D  G VF       Q              S |
| 8 | 2P42 | V                G    L          KG       D       T |
| 9 | 2X6M | G  V  S         R     G  R A    D      E          I |
| 10 | 3CFI | P        S V    GL W SG S TAP        IL      R       R |
| 11 | 3G9A | D    S        A      C L SN T  G       D      VN  S    R      K |
| 12 | 3JBC | S  T            G  GA       Q K            I |
| 13 | 3K1K | V  A  P      R Y    W  GSS E       D R |
| 14 | 3K74 | P      Y V R    GL W SM K  E        L      TS        K |
| 15 | 3K7U | T            LF    N      R  T |
| 16 | 3V0A | V    P        S V    EGF W SSAWDG  A    T  D    L    SN Q    G |
| 17 | 3ZKQ | P      T V    G  W ST T       R  L          R P |
| 18 | 4C58 | S      G      A  G  CS  S   R    Q   T AF L S      I  A |
| 19 | 4GRWf | E  V    P      I    G SC       ES |
| 20 | 4HEM | V        P    T  RN          NM |
| 21 | 4HEP | D  V    P    E  I    G SY  V  T    S        L      K  L |
| 22 | 4I0C | S      E  A    G  P       Q    RM      E    M |
| 23 | 4I13 | A      I      V GE  E    I   MN   V R    N    R |
| 24 | 4KML | P        G    SSD T      M  N      T |
| 25 | 4KRM | K E  S  T   T        SG  G        D      I |
| 26 | 4LAJ | V    P      I    G SC              Y K |
| 27 | 4LGP | V T    P  T   G I    H WLVC V V    A      L    D GI |
| 28 | 4LGS | V      S    A    S  L        AL    N |
| 29 | 4LHJ | V        Y    L  TSN      G    S |
| 30 | 4MQS | D      I    Q  G SC I   A    S  E           V |
| 31 | 4OCL | P      VD  A      T        R          R |
| 32 | 4QGY | V          I    G SC  P    A S K   M            K |
| 33 | 4QKX | Y    Q  L   N  N       A |
| 34 | 4S10 | S  R   A    K      S   DN N D |
| 35 | 4W6W | S        G  CVN    Q S  K L  E      L S |
| 36 | 4W6X | S      T    G  CS     Q   D  F R      I |
| 37 | 4W6Y | S        A   V   G  S        S        T |
| 38 | 4WEM | E        Q    GYLN G F        SN  S   G  F    K |
| 39 | 4WEN | P      T    Y    SK    H EF       T  D            K |
| 40 | 4WEU | K   Y      IA        L    D |
| 41 | 4X7C | D  V    P        Y    Q  L  S N      G            R R |
| 42 | 5IVN | V    P  T   T  I    G SC      FMN   D          I |

**Figure 4.** A pre-mutagenesis framework sequence variability across unstable nanobodies.

The online version of this article includes the following source data for figure 4:

**Source data 1.** Editable version of *Figure 4*.

Legend:
- unstable
- partially stable
- stable

IMGT# reference sequence:

```
IMGT#                                                                 1111111111111111
                11111111112222222234444444444455555556666777777777788888888888999999999900000011222222222
            123456789123456789012345690123456789012345678901234567890123456789012345678901234890123456 78
MUT#        MAQVQLQESGGGLVQAGGSLRLSCAASMGWFRQAPGKEREFVAATYYADSVKGRFTISRDNAKNTVYLQMNSLKPEDTAVYYCWGQGTQVTVSS
```

| MUT# | ID | Mutations (stable = green) |
|---|---|---|
| 7 | 1KXQ | V, V, *G*, I |
| 6 | 1KXT | V, Y, C L, RAN |
| 11 | 1KXV | V, S Y, G, G T V, M |
| 4 | 2X6M | V, R, *G* R A, I |
| 8 | 3CFI | P, S, G W G S T |
| 9 | 3G9A | A, C L N T, R |
| 4 | 3K1K | V P, R Y, W GSS E |
| 9 | 3K74 | P, Y, G W M K |
| 13 | 3V0A | V P, S, G W SAW G |
| 6 | 3ZKQ | P, T, G W T T |
| 4 | 4GRFf | V P, I, *G* C |
| 8 | 4HEP | V P, E I, *G* Y |
| 9 | 4I13 | I, V G, N |
| 2 | 4KML | P, G, SSD T, T |
| 7 | 4KRM | G G, I |
| 3 | 4LGS | V, A, S L, A |
| 2 | 4LHJ | V, Y, L TSN |
| 7 | 4MQS | I, *G* C I |
| 4 | 4OCL | V A, T |
| 5 | 4QGY | V, I, *G* C P, M |
| 2 | 4QKX | Y, Q L N |
| 7 | 4S10 | S R |
| 8 | 4WEM | E, Q, YLN G |
| 5 | 4WEN | P, T Y, K, H E |
| 4 | 4WEU | K Y |
| 4 | 4X7C | V P, Y, Q L S N |

| MUT# | ID | Mutations (partially stable = orange) |
|---|---|---|
| 3 | 1G6V | V, *G* T G, I |
| 9 | 1RJC | *G* V A, M |
| 5 | 1ZV5 | V, I, D *G* F, S |
| 4 | 3J6A | *G* GA, I |
| 3 | 4I0C | E A, *G* P, R, M |
| 6 | 4W6W | *G* CVN, *E*, L |

| MUT# | ID | Mutations (unstable = pink) |
|---|---|---|
| 5 | 1JTP | *G*, *L*, I |
| 3 | 2P42 | V, *G* L, T |
| 4 | 3K7U | LF |
| 9 | 4C58 | G, *G* CS S, *L*, I |
| 4 | 4HEM | V, T R |
| 3 | 4LAJ | V P, I, *G* C |
| 10 | 4LGP | V P, I, W C V V, I |
| 4 | 4W6X | T, *G* CS, *R*, I |
| 2 | 4W6Y | A V, *G* S, T |
| 6 | 5IVN | V P, T I, *G* C, I |

**Figure 5.** B sequence and stability variability across nanobodies following stage 1 mutagenesis *52Gly and 90X are bolded and italicized. The number of mutations made to each nanobody is reported in the left-most column.

The online version of this article includes the following source data for figure 5:

**Source data 1.** Editable version of *Figure 5*.

**Figure 6.** C sequence and stability variability across nanobodies following stage 2 mutagenesis (+G52F+X90Q).

The online version of this article includes the following source data for figure 6:

**Source data 1.** Editable version of *Figure 6*.

that mutation of non-interacting residues will generate conformational changes that impact target binding cannot be ruled out, this finding suggests that mutating nanobodies to adhere to our partial consensus framework likely minimizes impact on target binding, compared to strategies that rely on grafting variable loops onto established nanobody frameworks (*Saerens et al., 2005*). In order to both confirm target binding for our mutationally stabilized nanobodies, and to investigate the importance of maintaining framework residue identity in highly variable positions, we decided to compare our partial consensus framework stabilization approach to a full-consensus approach, in which all framework positions (excluding cysteines) are mutated to adhere to our stable consensus sequence.

We chose six mutationally stabilized nanobodies whose targets are easily expressed intracellularly (three targets, two nanobodies per target) to test for intracellular target binding (*Figure 8*). For each nanobody, wild-type, partial consensus (PC), and full-consensus (FC) variants were tested. Each nanobody was C-terminally linked with an FP, and each target was engineered to contain an N-terminal nuclear localization sequence (NLS). Wild-type and mutant nanobodies were co-transfected with either an off-target NLS construct, or NLS target into 293T and HeLa cells. H2B-mCherry was used as a nuclear marker to assess nuclear localization of nanobody signals. Target binding was evaluated based on observed nuclear localization of nanobody-FP signals in the presence of NLS target (*Figure 8A*), as well as pixel intensity measurements of captured fluorescence images (*Figure 8—figure supplement 1B, C*).

All six mutants stabilized through partial consensus mutagenesis were able to bind target (*Figure 8A* and *Figure 8—figure supplement 1C*). We were particularly interested to observe target binding for 3G9A, which has a large CDR3 loop that is normally anchored by an extra disulfide bond to FR2, assumed to be lost when translated in the cytoplasm. 4/6 wild-type nanobodies exhibited increased signal when coexpressed with target (*Figure 8—figure supplement 1B*), highlighting the ability of even poorly expressed nanobodies to bind an overexpressed target, albeit at lower levels when compared to their stabilized counterparts. Two of these nanobodies, 3K74 and 4I13, exhibited robust target binding in both their wild-type and mutationally stabilized forms. This suggested that the presence of their target, *E. coli* dihydrofolate reductase (DHFR), had a largely stabilizing effect.

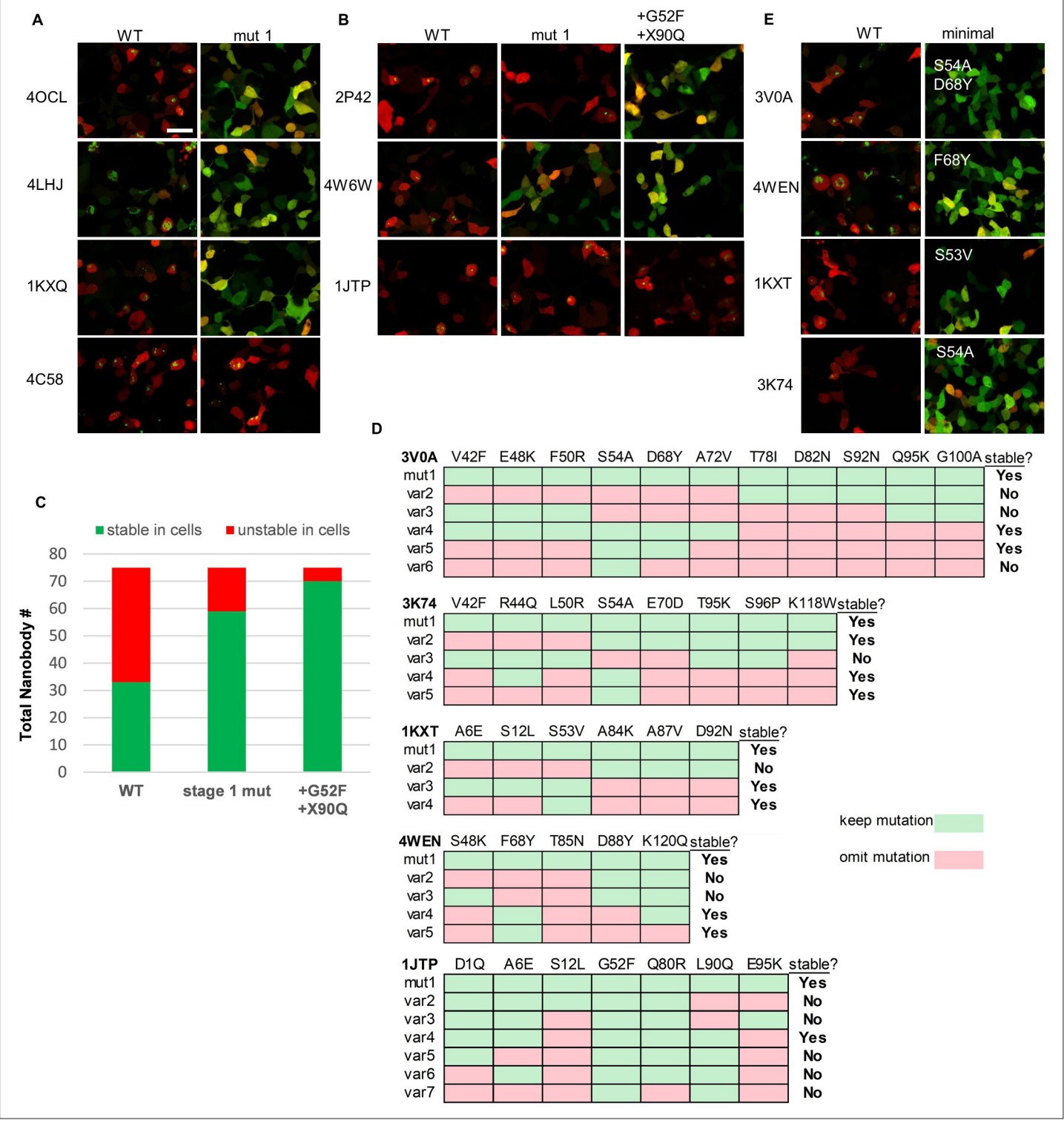

**Figure 7.** Mutationally stabilized nanobodies and specific stability drivers. (**A**) Examples of nanobody-TagBFP expression in 293T cells following transient transfection before (WT) and after (mut1) stage 1, conservation-based mutagenesis. Red signal is from co-transfected CAG-dsRed plasmid. (**B**) Examples of nanobody-TagBFP expression in 293T cells following transient transfection for nanobodies with Gly52. Wild-type, first-pass conservation-based mutants (mut1), and mutants with up to two additional mutations, Gly52Phe and X90Gln, are depicted. (**C**) Numbers of stable versus unstable nanobodies (75 total) following first-pass conservation-based mutagenesis (stage 1 mut) and final mutagenesis (+G52F+X90Q). (**D**) Example mutant variants tested for specific mutationally stabilized nanobodies in order to identify necessary stability mutations. (**E**) Examples of nanobody-TagBFP expression in 293T cells following transient transfection for nanobodies effectively stabilized by one to two mutations. Scale bar is 20 µm.

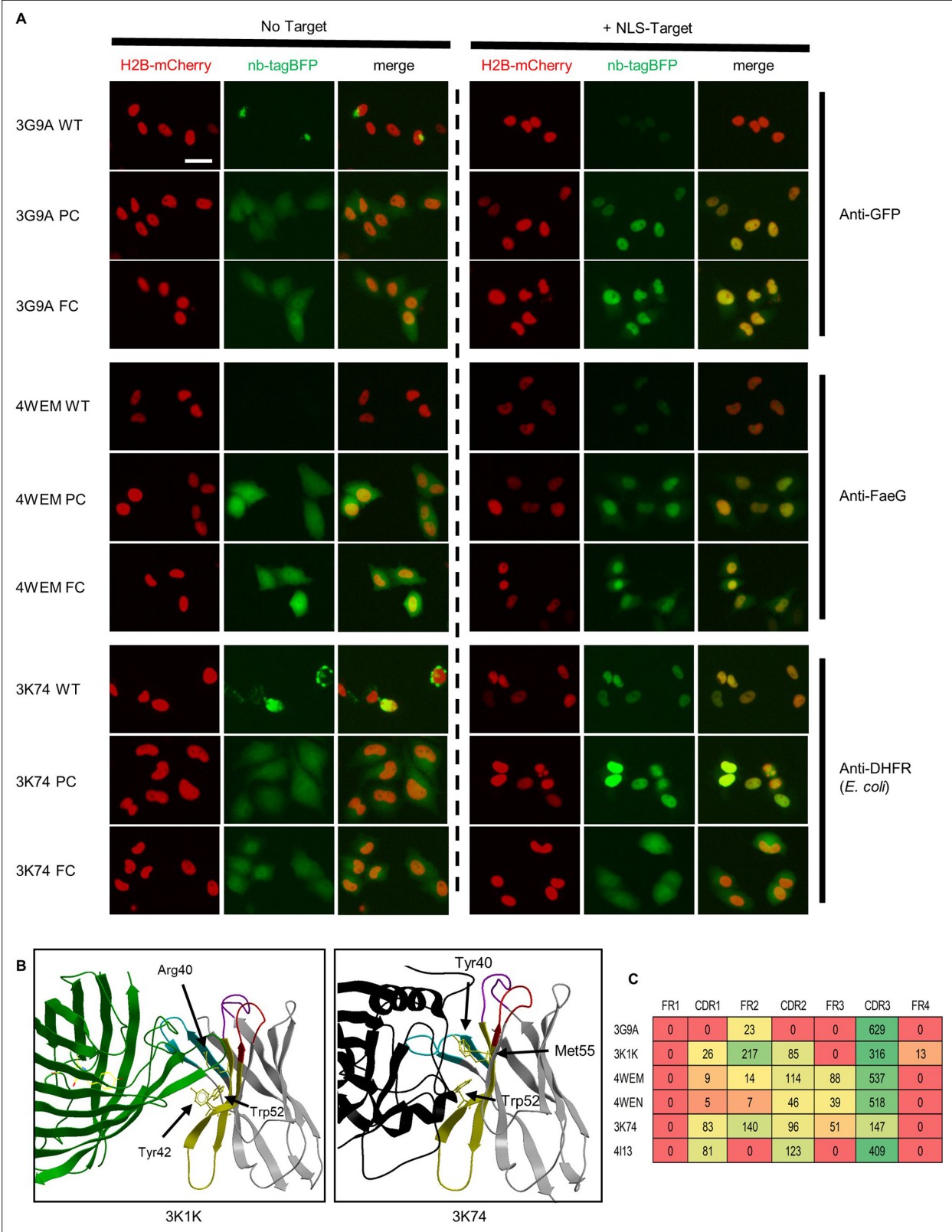

**Figure 8.** Target binding of parent and mutationally stabilized nanobodies. (**A**) Representative images of nanobody-TagBFP expression in HeLa cells in the presence and absence of nuclearly localized target. Wild-type (WT) nanobodies, partial consensus (PC) mutants, and full-consensus (FC) mutants are depicted. Red nuclear signal is from co-transfected CAG-H2B-mCherry plasmid. Transfected DNA and nuclear protein amounts were standardized by addition of off-target nuclear localization sequence (NLS) plasmid to transfection mix for the 'no target' condition. Scale bar is 25 µm. (**B**) Crystal

*Figure 8 continued on next page*

*Figure 8 continued*

structures of two nanobodies that lose target binding when mutated to adhere to a full-consensus framework are shown. Non-consensus framework residues directly contributing to target interface are depicted. (**C**) Target-interfacing surface area values in square angstroms (rounded to whole numbers) across distinct regions for nanobodies tested for target binding are shown. Values are taken from buried surface area interface values made available through PDBE-PISA.

The online version of this article includes the following figure supplement(s) for figure 8:

**Figure supplement 1.** Positional interactions and intensity measurements.

Our laboratory has previously described nanobody conditional stability in an engineered context (*Tang et al., 2016*), and it is expected for a subset of nanobodies to exhibit this target-dependent stability naturally.

Full-consensus mutagenesis, while stabilizing (*Figure 8A* and *Figure 8—figure supplement 1B*), resulted in significant reduction of nuclear nanobody signal fraction in the presence of NLS target for 5/6 tested nanobodies, suggestive of a reduction in target binding. Two of these nanobodies, 3K1K and 3K74, normally engage their targets along a concave framework interface (*Figure 8B*). Specifically, residues across FR2 form direct contacts with target, representing a common mode of nanobody-target binding (*Figure 8B, C*).

We next turned our attention to a nanobody with an atypical mode of target binding. Nanobody 5IVN (also known as nanobody BC2) targets a short N-terminal stretch of β-catenin, representing a rare epitope: a short, linear peptide amenable for use as an affinity tag. Corroborating previous findings, we found 5IVN to be unstable when expressed in cells (*Traenkle et al., 2015*). Unfortunately, while our mutagenesis approach stabilized 5IVN for cellular expression, it did not facilitate intracellular binding of its epitope as assessed by lack of co-localization with its target following cellular co-transfection of 5IVO-TagBFP and NLS-mCherry-epitope (data not shown). Binding was not achieved after limiting stabilizing mutagenesis to the single, necessary mutation to achieve intracellular stability, Ser54Ala. We suspect this lack of binding relates to loss of a crucial extra disulfide bond that normally orients its highly conformationally precise paratope, but that does not form intracellularly. This disulfide bond has previously been shown to be required for target binding (*Braun et al., 2016*).

Taken together, these target-binding experiments suggest that nanobodies stabilized by the partial consensus-based mutagenesis approach developed here are likely to retain target-binding function in cells, although certain conformational paratopes may not be amenable for intracellular recapitulation.

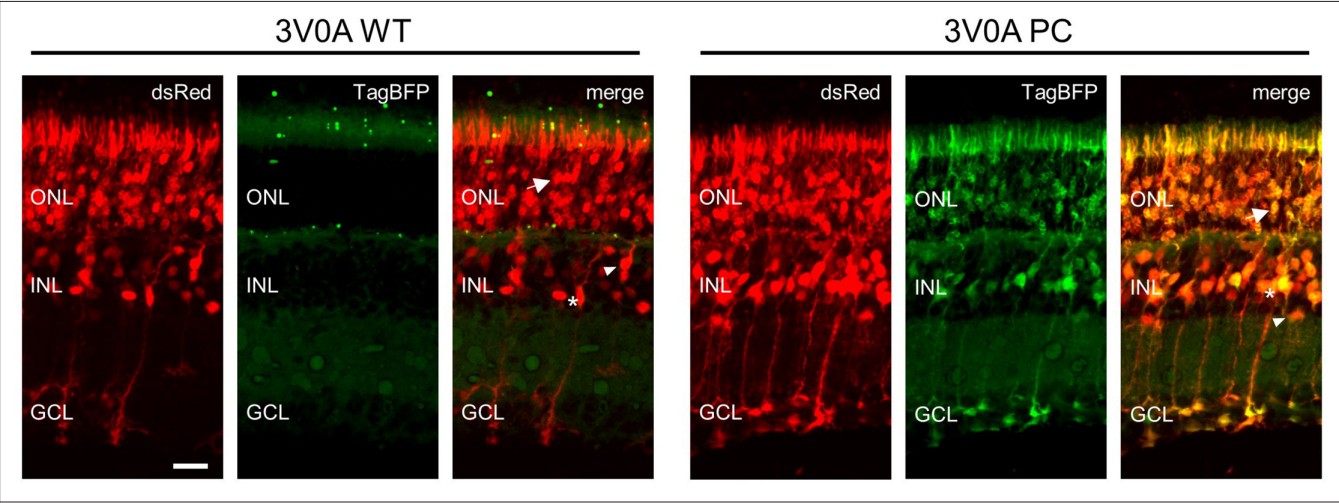

**Figure 9.** Nanobody expression in the murine retina representative Images of murine retina co-electroporated with CAG-dsRed and either wild-type or mutant CAG-3V0A-TagBFP plasmid. Retinas were electroporated on postnatal day 2 and harvested on postnatal day 12. Multiple cell types show expression following electroporation: photoreceptors (arrow), bipolar interneurons (triangle), and Mueller glia (asterisk) are noted on merged images. Scale bar is 20 μm.

## Nanobody intracellular stability in vivo

As many research and therapeutic applications of nanobody technology will require expression in vivo, we wanted to test whether improved nanobody intracellular stability established in cell lines would apply to in vivo settings. Electroporation of the retina in newborn mice is a means to deliver nanobody expression vectors to multiple cell types (*Tang et al., 2013*). Plasmids encoding wild-type or partial consensus mutant 3V0A-TagBFP were injected into the subretinal space of postnatal day 2 (P2) mice, along with a dsRed control plasmid, and electroporation was carried out. Tissue was harvested at P12, and was fixed, sectioned, and imaged. Representative images are displayed in *Figure 9*. Red fluorescent signal from the dsRed control plasmid delineated areas of successful electroporation. Little to no soluble TagBFP signal was observed in cells electroporated with wild-type 3V0A-TagBFP. However, strong TagBFP signal was observed in cells electroporated with partial consensus mutant 3V0A-TagBFP, mirroring observations in cultured cells. These cells included highly specialized sensory neurons (rod photoreceptors), interneurons (bipolar cells), and glia (Mueller glia), suggesting that mutationally stabilized nanobodies are stable in multiple, distinct cell types.

We additionally wanted to validate that a nanobody mutationally stabilized by our partial consensus approach was capable of binding an endogenous target in vivo. Although nanobodies that target intracellular, endogenous factors are currently limited, studies engaging, perturbing, and reporting on endogenous factors in live organisms will ultimately help differentiate nanobody technology from conventional antibody reagents. Fortunately, we had in our collection a nanobody, NBE9 (*Li et al., 2012*), that recognizes an endogenous protein, glial fibrillary acidic protein (GFAP), present in retinal Mueller glial cells. It was unstable initially and was stabilized through mutagenesis. Plasmids encoding wild-type and stabilized variants were electroporated into the rd1 mouse model of retinal degeneration at P2 as TagBFP fusions, along with a CAG-dsRed co-electroporation control. In the rd1 model, rod photoreceptors progressively degenerate starting around postnatal day 8 (*Chang et al., 2007*), and GFAP expression is upregulated in Mueller glia (*Goel and Dhingra, 2012*). GFAP staining with conventional antibodies has shown that GFAP increases in filaments within radial processes, as well as in Mueller glial endfeet, located within the ganglion cell layer (GCL) (*Goel and Dhingra, 2012*). GFAP is additionally expressed by astrocytes in the GCL (*Goel and Dhingra, 2012*), but astrocytes in the retina are not electroporated using our protocol (*Matsuda and Cepko, 2004*). Representative images from these experiments are shown in *Figure 10*.

As a control for the distribution of a well-expressed, stable nanobody without a target, we used 3V0A PC. The signal of this nanobody was predicted to match the diffuse signal of the dsRed encoded by the co-electroporation control. We did indeed observe this, as mutationally stabilized 3V0A-TagBFP signal matched that of dsRed in all electroporated cell types, which included Mueller glia (*Figure 10A*). To determine if NBE9-TagBFP signal colocalized with GFAP, sections were stained with an anti-GFAP conventional antibody. In keeping with its instability, WT NBE9 signal was almost completely absent within electroporated retinas. The only signal observed was in spherical accumulations in the GCL, where it colocalized with accumulated GFAP protein (*Figure 10B*). A NBE9 variant with the single, strongly stabilizing mutation S54A, both rescued expression of the nanobody and resulted in a striking expansion of the signal seen in the spherical accumulations of GFAP within the GCL (*Figure 10C*). The NBE9 PC mutant exhibited less of these colocalizing accumulations than the S54A mutant (*Figure 10D*). It did, however, track with some radial, GFAP-positive processes somewhat distinctly from the more diffuse dsRed signal. Taken together, both wild-type and mutationally stabilized NBE9 appeared to colocalize with GFAP in Mueller glial endfeet, with variants differing in expression level and, perhaps, propensity to form aggregates with GFAP.

To further explore potential applications of partial consensus stabilizing mutagenesis, we tested mutant nanobody expression in bacteria. Several groups have illustrated powerful applications taking advantage of nanobodies expressed in bacteria (*Harmsen et al., 2006*; *Vandenbroucke et al., 2010*; *Chowdhury et al., 2019*). Bacteria offer a therapeutic modality, as they can be delivered as factories for nanobody production. Moreover, stable expression of nanobodies in bacteria would facilitate intrabacterial studies, and may contribute to improved production yields for nanobody reagents produced in bacteria. Five mutationally stabilized nanobodies and their parents were tested for expression in BL21 *E. coli*. Nanobodies were fused C-terminally to mCherry and expression was assessed by fluorescence microscopy. All five wild-type nanobodies exhibited hallmarks of instability in bacteria (*Figure 11*). 4/5 wild-type nanobodies exhibited protein aggregation as illustrated by focal inclusion

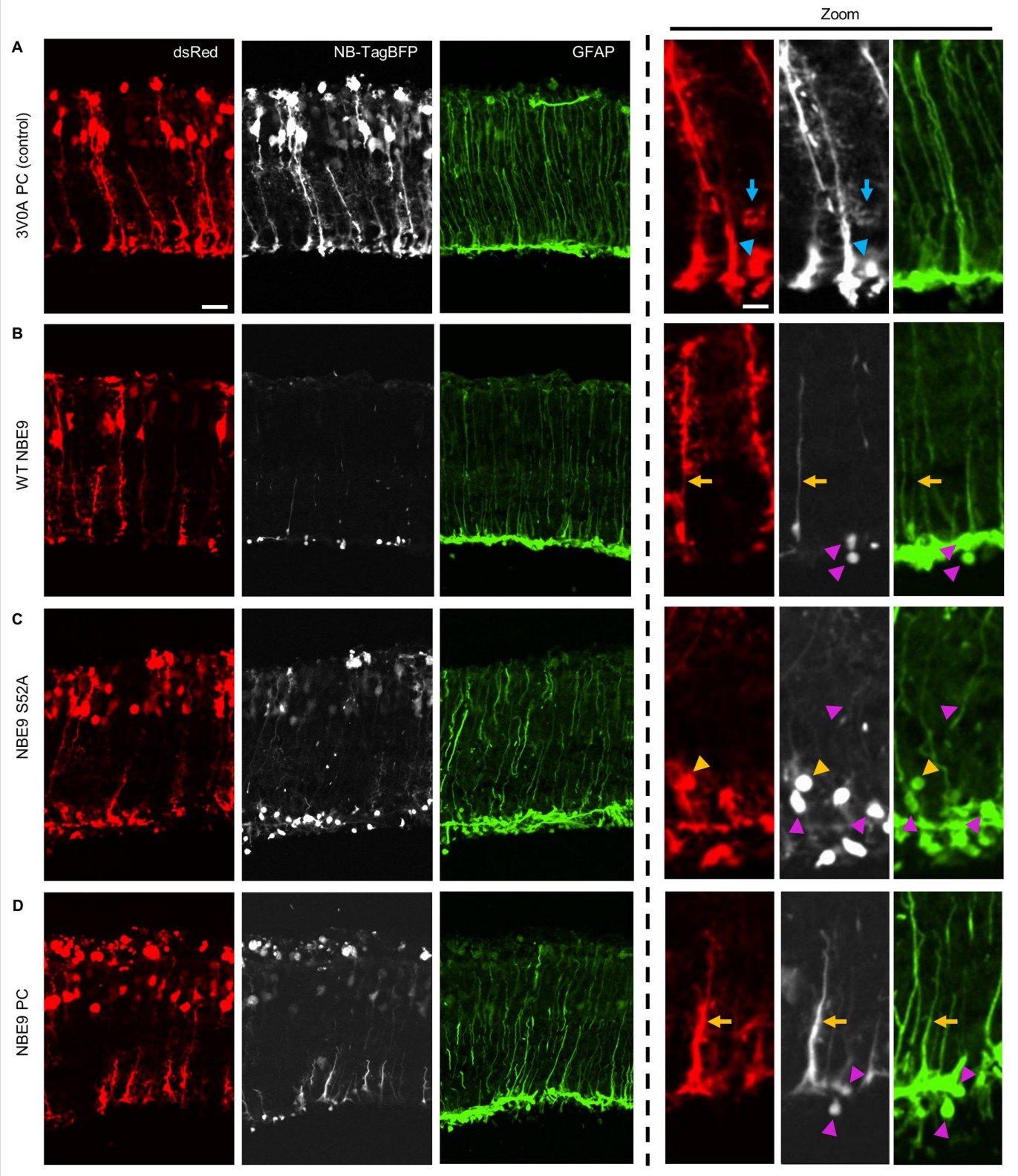

**Figure 10.** Nanobody targeting endogenous factor in the murine retina. (**A–D**) Representative Images of rd1 murine retina co-electroporated with CAG-dsRed and CAG-nanobody-TagBFP plasmids. NBE9 is a nanobody that targets glial fibrillary acidic protein (GFAP), a protein whose expression in Mueller glia is upregulated in the rd1 retinal degeneration model. Retinas were electroporated on postnatal day 2 and harvested on postnatal day 20. Triangles denote features of accumulated protein, and arrows denote radial processes. Blue indicators denote features observed in both the dsRed and

*Figure 10 continued on next page*

*Figure 10 continued*

nanobody-TagBFP channels, pink indicators denote features observed only in the nanobody-TagBFP and GFAP channels, and orange indicators denote features observable across all three channels. Scale bars for main panels are 20 μm and for higher magnification panels are 4 μm.

bodies, in addition to diminished cytoplasmic fluorescence. 3V0A did not appear to aggregate, but exhibited low fluorescent signal. All five mutationally stabilized nanobody variants exhibited a marked increase in fluorescence level. 1ZV5, 4WEN, and 1RJC mutants showed no signs of protein aggregation, in contrast to their wild-type parents. Mutant 3G9A exhibited the most intense fluorescent signal, but also exhibited fluorescent inclusion bodies, possibly due to higher nanobody concentration.

## Discussion

Here, we describe a general method for modifying nanobody sequences to facilitate stable intracellular expression. A partial consensus framework was distilled from the most highly conserved positional residues across a large group of intracellularly stable nanobodies. This framework was applied broadly across unstable nanobodies to rescue intracellular stability in a great majority of cases. As new nanobodies are generated against intracellular targets, this stabilization approach should prove effective in achieving stable expression while maintaining target binding.

Consensus-based stabilization has been applied to antibody fragments in the past; Pluckthun and colleagues were the first to investigate stability and solubility differences between the consensus sequences of distinct families of human immunoglobulin variable domains in non-reducing in vitro conditions, as well as differences in expression in non-reducing bacterial periplasm (***Knappik et al., 2000***; ***Ewert et al., 2003***). They identified positional sequence differences between well- and poorly-expressed consensus frameworks that impacted charged interactions and hydrophobic packing of upper and lower core hydrophobic residues. Although this work focused on domain stability in a context facilitating disulfide bond formation, it highlighted an important observation: while sequence changes between variable domains did not always yield significant differences in in vitro stability and solubility measurements, soluble yield and aggregation of expressed fragments in bacteria could vary dramatically, suggesting folding dynamics independent of domain stability.

Steipe and Wirtz applied consensus framework mutagenesis to $V_L$ and $V_H$ domains to enhance intracellular stability for conventional $F_V$ fragments (***Ohage et al., 1999***; ***Wirtz and Steipe, 1999***). This approach was successful as conventional antibodies are amenable for loop grafting onto full donor frameworks due to their near exclusive reliance on CDRs for binding. Our partial consensus approach takes into account variable binding modes observed across nanobody crystal structures. Nanobodies

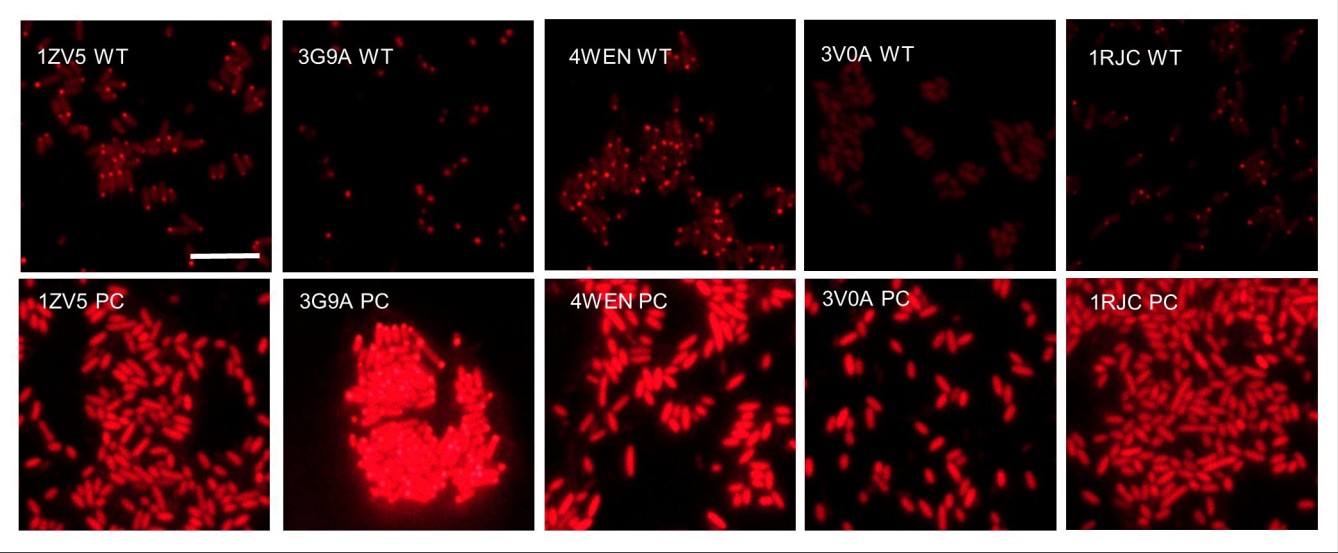

**Figure 11.** Nanobody expression in *E. coli* representative images of BL21 *E. coli* transformed with either wild-type or mutant pRsetB-nanobody-mCherry plasmid. Images were taken following 4 hr of protein induction with 250 μM Isopropyl β-D-1 thiogalactopyranoside(IPTG). Scale bar is 10 μm.

often rely on direct framework engagement for target binding. We show that positional sequence variability across nanobody frameworks correlates with target interface formation, and that maintaining framework positions of high sequence variability is important for preserving target binding in some, perhaps many, cases.

While investigating intracellular target binding, we noted that some nanobodies were able to effectively bind their targets despite their observed instability when target is absent. Such nanobodies may be useful reagents without the need of stability engineering in certain cases. However, in cases of high nanobody expression and low target availability, an unstable, aggregation-prone fraction of unbound nanobody may prove problematic for studies concerning subcellular protein localization, or for reasons of cellular toxicity. Indeed, one potential therapeutic application for intracellular nanobodies is to block protein aggregation that leads to various neurodegenerative diseases (*Messer and Joshi, 2013*). Several groups have investigated nanobodies as potential reagents for blocking aggregation of a range of cytotoxic, aggregation-prone factors including prion protein, amyloid beta, and alpha synuclein (*Abskharon et al., 2014*; *Kasturirangan et al., 2010*; *Chatterjee et al., 2018*). In these contexts, nanobody stability and solubility may prove critical to ensuring that no undue cellular stress is imparted by the treatment agent.

Additionally, the ability to tweak nanobody framework properties allows for the generation of a range of reagents that behave differently in an intracellular context, with each reagent potentially useful for different applications. We delivered three versions of NBE9, a nanobody that binds GFAP, in vivo in the murine retina, with the primary goal of evaluating the ability to bind target. We observed three versions, all capable of binding to GFAP, but each exhibiting a different expression signature, possibly due to a combination of factors including intracellular stability, aggregation propensity, and cellular context. Wild-type NBE9 showed sparse labeling of accumulated GFAP in Mueller glial endfeet, with little protein in cell bodies or processes. The single mutant, S54A, resulted in higher expression and seemed to encourage the formation of spherical GFAP aggregates in Mueller glial endfeet, where it primarily localized. NBE9 PC also exhibited higher expression, but was more distributed throughout the cell, including within radial, GFAP+ processes. Interestingly, while NBE9 PC exhibited GFAP-binding capability, it did not seem to encourage GFAP aggregation to the same extent as the S54A single mutant. It is possible that a difference in binding affinity to GFAP underlies this observation, and/or that a difference in aggregation propensity of the two variants influences GFAP dynamics. Depending on the intended application, sparse labeling, impact on target localization, or broader nanobody distribution may all be desirable properties.

The potential for intracellular deployment of nanobody-based reagents is still far from being realized. Taking full advantage of the modularity and versatility inherent of these pared-down structures will require methods such as those proposed here to condition nanobodies for the intracellular environment.

## Materials and methods
### Compilation of nanobody sequence and structure data
Nanobody sequences were pulled directly from the PDB database (sequences for this study compiled in 2016). To generate an interface atlas profiling positions of interaction between each nanobody and its target (linked data provided), interface data were referenced from the Proteins, Interfaces, Structures, and Assemblies tool (PDBePISA), provided by the European Bioinformatics Institute (EMBL-EBI). Values for positional interactions were taken from Buried Surface Area values (Å$^2$) for each residue.

### Generation and cloning of nanobody sequences
Nanobody sequences were synthesized as double stranded DNA fragments (gBlocks) by IDT, and cloned directly into a CAG expression plasmid (Addgene plasmid 11150) (*Matsuda and Cepko, 2004*) in frame and linked with TagBFP via Gibson assembly (New England Biolabs, E2611). For bacterial expression constructs, nanobody sequences were PCR amplified with primers containing terminal regions of homology for cloning into a pRset plasmid (Addgene plasmid 3991) (Invitrogen, V35120) in frame and linked to mCherry. DH5α *E. coli* were transformed with assembled DNA and cultured in 4 ml cultures overnight in LB medium with 100 µg/ml carbenicillin. Plasmid DNA was purified using Qiagen miniprep kits.

Mutant nanobody sequences were designed by comparing intracellularly unstable nanobodies to a partial consensus framework sequence of stable nanobodies, and changing amino acids in each unstable nanobody to match the partial consensus sequence at each position. Nanobody framework regions were defined by The International Immunogenetics Information System (IMGT), a global reference for immunogenetics. The consensus sequence for stable nanobodies was generated by calculating the most frequent amino acid at each framework position across 33 intracellularly stable nanobodies. The partial consensus sequence represents the subset of consensus positions at which 80% or greater of the 33 stable nanobodies had the same amino acid (*Figure 10B*).

## Cell culture and transfection

All constructs used in mammalian cell transfection experiments were cloned into the pCAG plasmid. Plasmid DNAs encoding nanobody-TagBFP sequences were transfected into both 293T cells and HeLa cells. Experiments were first conducted in 293T cells, tested across three replicates, and later tested in HeLa cells. One day prior to plasmid transfection, cells were plated in black, clear bottomed 96-well plates (CLS3603, Sigma-Aldrich) at roughly 10,000 cells/well and incubated at 37°C, and 5% $CO_2$. For transfection of cells in each well, 5 µl of serum-free media was added to 200 ng of plasmid DNA (100 ng CAG-nanobody-TagBFP plasmid and 100 ng CAG-dsRed plasmid [Addgene plasmid 11151], *Matsuda and Cepko, 2004* or CAG-H2B-mCherry plasmid). For target-binding experiments, an additional 100 ng of NLS target expressing plasmid or 100 ng of an off-target, control NLS expressing construct was added. 1 µl (or 1.5 µl for target-binding experiments) of 1 mg/ml PEI (Polysciences, 24765-2) dissolved in water was added to DNA in media and vortexed for 10 s. Resulting transfection mix was left to sit at room temperature for 10 min before being added to cells.

## Live fluorescent imaging and stability scoring

For initial stability scoring, 75 nanobodies were transfected into HEK293T cells (as described above) as TagBFP fusions driven by a CAG promoter. Roughly 24 hr after transfection, TagBFP signal was evaluated using a Leica DMI3000B microscope and a ×20 objective lens. CAG-dsRed signal served as both a transfection control and orienting signal to assess cellular morphology. The 75 nanobodies were binned broadly into 'stable' and 'unstable' groupings based on the observed character of the TagBFP signal for each nanobody. Nanobodies with TagBFP signal that filled cells, mirroring dsRed signal, and that exhibited, at most, only minor and infrequent fluorescent puncta, were binned into the 'stable' group. Nanobodies with sparse to absent TagBFP signal, and/or that exhibited major and frequent fluorescent puncta, were binned into the 'unstable' group (*Figure 1A*). Groupings were validated across three separate rounds of transfection and evaluation in HEK293T cells, and further validated in HeLa cells. Grouping criteria were assigned to capture major differences in intracellular expression between groups, rather than subtle differences in expression pattern or character within groups.

For target-binding experiments in HeLa cells, images were taken using the same Leica DMI3000B microscope and ×20 objective lens roughly 48 hr after transfection. Excitation time was kept consistent between samples.

Imaging of parent and mutant nanobodies to evaluate mutational stabilization was carried out via automated confocal imaging. Using a PE Opera high-throughput confocal imaging system (https://www.flyrnai.org/supplement/BRO_OperaHighContentScreeningSystem.pdf) provided by the *Drosophila* RNAi Screening Center (https://fgr.hms.harvard.edu/), live, transfected HEK293T cells were imaged in 96-well plates using a water-immersion ×40 objective. Ten fields for image acquisition, standardized across each well, were assigned prior to imaging. Through an automated protocol, six Z stacks were taken in each field in both red and blue channels, spanning 12 µm. Following image acquisition, max-projection images were generated, and nanobody-TagBFP signal was evaluated.

## Retinal electroporation

Wild-type and mutant CAG-nanobody-TagBFP plasmids were each electroporated along with control CAG-dsRed plasmid (1.5 µg/ml total DNA) into the retina of P2 mouse pups. For electroporations to look at endogenous GFAP engagement, the rd1 mouse model of retinal degeneration was used (*Chang et al., 2007*). For each condition, three pups received wild-type nanobody plasmid, and three

pups received mutant nanobody plasmid. Electroporation was executed according to the methodology described in *Matsuda and Cepko, 2004*.

## Retinal histology and imaging

Electroporated retinas were harvested at P12 (P20 for rd1 mice). Retinas were fixed in 4% formaldehyde for 30 min, transferred to phosphate-buffered saline (PBS) for 10 min, and transferred to 15% sucrose in PBS for 30 min, all at room temperature. Retinas were then embedded in OCT, flash frozen, and stored at −80°C. 12 µm sections were made using a cryostat, placed on glass slides, and mounted with Fluoromount-G mounting media (Thermo Fisher, 00-4958-02). Retinas from rd1 mice were stained for GFAP using chicken anti GFAP primary antibody (1:3000, Aves Labs, cat.#: GFAP) and secondary antibody conjugated to AlexaFluor 488 (Jackson ImmunoResearch). Images were taken using a Nikon Ti2 inverted microscope (spinning disk confocal) with a ×40 oil immersion objective lens.

## Bacterial expression and imaging

Nanobody sequences were cloned into the bacterial expression vector pRsetB in frame and fused to mCherry. Plasmid was transformed into BL21(DE3) *E. coli* (New England Biolabs, C2527), and individual colonies were picked into 5 ml LB and cultured overnight at 37°C and shaken at 250 RPM. In the morning, 1 ml of each culture was added to 4 ml of M9 minimal media, and IPTG was added to a final concentration of 250 µM to induce protein expression. Induction cultures were incubated for 4 hr at 37°C and 250 RPM. 40 µl of induced culture was pipetted onto 3% M9-agar on a glass slide and covered with a glass coverslip. Bacteria were imaged with a Nikon Eclipse e1000 microscope using a ×60 oil immersion objective lens. All images were taken with consistent excitation time. Experiments were performed in triplicate.

## Image processing

Images were processed using ImageJ. Images from *Figures 1 and 7* received the following processing and no other adjustments: (1) images were converted to 8-bit, (2) Z stacks for each channel (red and blue) were merged to create maximum Z projections, and (3) maximum Z projections for each channel were merged with one another, with red images inserted into the red channel, and blue images inserted into the green channel for greater contrast. Retinal images in *Figure 9A* received the same processing, but received cropping and additional contrast adjustment (all images received consistent adjustment to contrast). Retinal images in *Figure 10* received the same processing, but with blue images inserted into the gray channel, and green images inserted into the green channel. Images from *Figures 8 and 9B* (without Z stacks) were similarly converted to 8-bit, were color-channel merged, and received equally administered contrast adjustment. Images from *Figure 8* were cropped.

## Image quantification

Pixel intensity quantification presented in *Figure 8—figure supplement 1B, C* were carried out in ImageJ, and measured across three uncropped images for each condition, taken from separate biological replicate experiments. Binary masks for H2B-mCherry and nanobody-TagBFP images were generated by converting images to 8-bit, enhancing contrast, and adjusting threshold using the standardized 'triangle' setting. Background was eliminated from otherwise unprocessed images via subtraction with generated binary masks. Normalized intensity measurements (*Figure 8—figure supplement 1B*) were generated by measuring total pixel values of each background-subtracted image, and calculating the ratio of nanobody (blue) to H2B-mCherry (red) signal for paired images.

For nuclear colocalization measurements (*Figure 8—figure supplement 1C*), cells lacking both markers needed to be excluded from analysis. This was achieved using binary masks and the morphological reconstruction function provided through the MorphoLibJ plugin (https://imagej.net/plugins/morpholibj), using H2B-mcherry signals as a marker and nanobody-TagBFP signals as a mask. Plugin results were validated manually to ensure cells were properly included/excluded. Measurement of total nanobody-TagBFP pixel values for background-subtracted images, and of pixel values colocalizing with paired H2B-mcherry binary masks gave the nuclear fraction of total nanobody-TagBFP pixel values for each image (+/−target).

# Additional information

### Funding

| Funder | Grant reference number | Author |
|---|---|---|
| Howard Hughes Medical Institute | | John G Dingus |
| Cure Huntington's Disease Initiative | | Constance L Cepko |
| National Eye Institute | K99EY032110 | Ryoji Amamoto |

The funders had no role in study design, data collection, and interpretation, or the decision to submit the work for publication.

### Author contributions

John G Dingus, Conceptualization, Data curation, Formal analysis, Validation, Investigation, Visualization, Methodology, Writing - original draft, Writing - review and editing; Jonathan CY Tang, Conceptualization, Data curation, Investigation, Methodology; Ryoji Amamoto, Grace K Wallick, Investigation; Constance L Cepko, Resources, Supervision, Funding acquisition, Project administration, Writing - review and editing

### Author ORCIDs

John G Dingus (iD) http://orcid.org/0000-0001-7928-8067
Jonathan CY Tang (iD) http://orcid.org/0000-0002-2376-6901
Constance L Cepko (iD) http://orcid.org/0000-0002-9945-6387

### Ethics

This study was performed in strict accord with the recommendations in the guide for the care and use of laboratory animals of the National Institutes of Health. All of the animals were handled according to approved institutional animal care and use committee protocol number ISI 001679 of Harvard University. Injections into neonatal eyes were performed under cryo-anesthesia, and all efforts were made to reduce pain and suffering, including use of an analgesic, buprenorphine. Animals were sacrificed using $CO_2$ fixation and cervical dislocation following the guidelines of the American Veterinary Medical Association for euthanasia.

### Decision letter and Author response

Decision letter https://doi.org/10.7554/eLife.68253.sa1
Author response https://doi.org/10.7554/eLife.68253.sa2

---

# Additional files

### Supplementary files

- Transparent reporting form
- Supplementary file 1. Full nanobody sequences and targets.
- Supplementary file 2. Nanobody interaction interfaces.

### Data availability

All data analyzed during the study are included in the manuscript and supporting files.

The following previously published dataset was used:

| Author(s) | Year | Dataset title | Dataset URL | Database and Identifier |
|---|---|---|---|---|
| Krissinel E, Henrick K | 2007 | Protein interfaces, surfaces and assemblies. European Bioinformatics Institute | http://www.ebi.ac.uk/pdbe/prot_int/pistart.html | Electron Microscopy Data Bank, prot_int/pistart |

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
