## [Editor Report]

This important study developed an innovative and powerful approach for improving the stability of nanobodies when expressed in an intracellular environment. The authors provide convincing evidence that by mutating key amino acids, they could enhance the stability of a majority of nanobodies in an intracellular environment without affecting their binding specificity. This study will be of general interest to a growing number of researchers using nanobodies as tools for their biological investigations.

---

## [Decision Letter]

**Decision letter after peer review:**

Thank you for submitting your article "A general approach for stabilizing nanobodies for intracellular expression" for consideration by *eLife*. Your article has been reviewed by 3 peer reviewers, and the evaluation has been overseen by a Reviewing Editor and Olga Boudker as the Senior Editor. The following individuals involved in review of your submission have agreed to reveal their identity: Aashish Manglik (Reviewer #1); James S Trimmer (Reviewer #2).

All reviewers appreciate the value of this study, and we invite you to submit a revision that will address reviewers' critiques. The reviewers have discussed their reviews with one another, and the Reviewing Editor has drafted this to help you prepare a revised submission.

Essential revisions:

1) The consensus after discussions among the reviewers is that the revised manuscript should include quantification of the expression levels, stability, and solubility of the nanobodies when expressed as intrabodies.

2) Demonstrating target binding inside the cells is retained in nanobody mutants that are enhanced in the above characteristics will be necessary. One specific suggestion is to use your nuclear localization assay for the 6 nanobodies to quantify the fraction of nanobody fluorescence that is nuclear localized in cells with or without nuclear-localized target expression, which will yield some insights as to the relative binding affinity of the various mutant isoforms.

3) Please also address the specific critiques of reviewers below as much as possible.

We look forward to a revised manuscript.

*Reviewer #1 (Recommendations for the authors):*

Several considerations may help clarify and put this work into broader context:

It is a bit unclear how the authors chose their set of nanobodies – currently there are more than 75 unique nanobodies in the PDB. Some of the nanobodies in the dataset are very closely related – was there any filtering done to remove nanobodies that are very similar in sequence?

It is also a bit unclear what species these nanobodies were derived from. Prior work has demonstrated that llama derived nanobodies are generally more stable than those derived from alpaca or camels. Is the consensus framework closest to the germline sequences of any specific animal? Does this reflect the bias of animals typically used for nanobody generation?

*Reviewer #2 (Recommendations for the authors):*

The impact statement should be revised to explicitly state that this study was limited to a small subset of existing nanobodies and may not broadly hold for all: "A majority of the set of nanobodies tested here are unstable when expressed intracellularly……"

The authors should somehow address, perhaps bioinformatically, whether the 75 nAbs they interrogated are representative of nanobodies in general. Perhaps these particular nanobodies were developed/screened/selected for their exceptional properties when expressed extracellularly, in that they represent those that are capable of forming high-quality crystals. At worst this could be a negative selection for those that function as intrabodies, or at least may be biased relative to nanobodies in general as to their utility as intrabodies.

How do the authors distinguish between nanobodies that are degraded versus those that are just not expressed? They say several replicates were performed, but were independent clones and/or plasmid preps tested?

Should cite extensive analogous work of Honniger and Pluckthun (and others) on consensus sequence-based stabilization of ScFVs.

As a final figure a simple flow chart or dichotomous key that researchers could use to triage their nanobodies would be extremely useful to researchers. This would distiill their recommendations into a simple form. For example,

1: is there a X residue at position Y, if yes mutagenize to Z. If no go to 2.

2: is there a XX residue at position YY, if so mutagenize to ZZ. If not go to 3. Etc. for all key elements.

*Reviewer #3 (Recommendations for the authors):*

1. The authors need to find a more quantitative and informative assay for measuring solubility and this needs to be measured while controlling for expression level.

2. Affinity needs to be measured in a quantitative and accurate manner using SPR or a similar method.

3. Affinity needs to be measured for all stabilized nanobodies and compared to the measurement before stabilization.

4. At least one of the stabilize nanobodies needs to be tested against its endogenous target to determine if the nanobody binds sufficiently to immunoprecipitate or colocalize with target.

---

## [Author Response]

Reviewer #1 (Recommendations for the authors):Several considerations may help clarify and put this work into broader context:It is a bit unclear how the authors chose their set of nanobodies – currently there are more than 75 unique nanobodies in the PDB. Some of the nanobodies in the dataset are very closely related – was there any filtering done to remove nanobodies that are very similar in sequence?

The 75 nanobodies were chosen based on our search of the PDB as of 2016, when this work began. It is certainly possible that our manual search missed a few structures, but we believe our list represents a great majority of available structures as of 2016. We did not filter our list with a strict homology cutoff, but excluded sequences that were identical or very nearly identical. There remain, however, a few nanobodies in our list that are closely related.

It is also a bit unclear what species these nanobodies were derived from. Prior work has demonstrated that llama derived nanobodies are generally more stable than those derived from alpaca or camels. Is the consensus framework closest to the germline sequences of any specific animal? Does this reflect the bias of animals typically used for nanobody generation?

The species of each nanobody is included in Table 1. As highlighted, a great majority (48/75) of the sequences in our list are llama sequences. This does, indeed, reflect the fact that the majority of nanobodies are currently derived from llama, and means that our consensus sequence most closely resembles llama. However, we have included a comparison between framework sequences of different species in our list in a supplemental figure.

Reviewer #2 (Recommendations for the authors):The impact statement should be revised to explicitly state that this study was limited to a small subset of existing nanobodies and may not broadly hold for all: "A majority of the set of nanobodies tested here are unstable when expressed intracellularly……"

We have revised this statement in order not to overgeneralize.

The authors should somehow address, perhaps bioinformatically, whether the 75 nAbs they interrogated are representative of nanobodies in general. Perhaps these particular nanobodies were developed/screened/selected for their exceptional properties when expressed extracellularly, in that they represent those that are capable of forming high-quality crystals. At worst this could be a negative selection for those that function as intrabodies, or at least may be biased relative to nanobodies in general as to their utility as intrabodies.

While we have not shown this bioinformatically, we believe that the nanobodies in our list cover a representative sequence space, especially of sequences originating in llama. Specifically, nanobody framework sequences, the sequence regions of focus for our manuscript, have been profiled previously, and their similarity to human type 3 VH domains has been noted^1^. Additionally, our framework sequences have excellent agreement with a consensus sequence derived from llama immune genes by McMahon and colleagues^2^.

How do the authors distinguish between nanobodies that are degraded versus those that are just not expressed? They say several replicates were performed, but were independent clones and/or plasmid preps tested?

In general, we did not test multiple independent clones/plasmid preps for every nanobody, although many nanobodies that were included in multiple experiments have been recloned and prepped several times with consistent results. All nanobody preps that were included in experiments were sequence verified, and our experience with cloning into standardized cloning vectors gives us confidence that our results reflect consistent expression levels.

However, one mistake must be addressed related to this comment. In our initial submission, plasmid preps of wild-type nanobodies that had been cloned several years prior (and sequence-verified at the time) were used in intracellular binding studies (represented in figure 4 and supplemental figure 1). After our initial submission, it was discovered that the quality of the DNA in some of these plasmid preps had deteriorated over time, resulting in inaccurate expression representation. The most drastically deteriorated sample was for wild type nanobody 3K1K (a GFP binder), which exhibits robust expression and binding ability in cultured HeLa cells, and only exhibits aggregation propensity at higher levels of expression (observed in HEK293T cells). Upon this discovery, binding experiments were re-done, with all nanobodies re-prepped and resequenced. New results better reflect the ability of even unstable nanobodies to bind targets in a co-expression context, albeit at lower overall levels.

Should cite extensive analogous work of Honniger and Pluckthun (and others) on consensus sequence-based stabilization of ScFVs.

Thank you for this comment. We now cite the work of Honniger and Plucktun related to consensus mutagenesis of scFvs.

As a final figure a simple flow chart or dichotomous key that researchers could use to triage their nanobodies would be extremely useful to researchers. This would distiill their recommendations into a simple form. For example,1: is there a X residue at position Y, if yes mutagenize to Z. If no go to 2.2: is there a XX residue at position YY, if so mutagenize to ZZ. If not go to 3. Etc. for all key elements.

Thank you for the suggestion. We now include a supplemental chart simplifying our recommendations.

Reviewer #3 (Recommendations for the authors):1. The authors need to find a more quantitative and informative assay for measuring solubility and this needs to be measured while controlling for expression level.

Our investigation is guided most directly by expression observed via fluorescence microscopy in live cells. Our study does not aim to uncover deep structural insights into nanobody folding in the intracellular environment, but rather took a broad approach in order to determine a generalizable method for improved expressibility of nanobodies. However, in this Revision, we have quantified nanobody localization in cells in response to nuclear target expression, as suggested by the Reviewers. We used wild type and mutated nanobodies, including partial and full consensus mutagenesis (see Figure 4). We included normalized nanobody expression measurements, based on analysis of pixel intensities from florescence microscopy images, for nanobodies in the presence and absence of target. We believe this gives a clearer picture of expression differences and nanobody-target colocalization for wild type and mutationally stabilized nanobodies.

2. Affinity needs to be measured in a quantitative and accurate manner using SPR or a similar method.

Please see response to recommendation #3 below.

3. Affinity needs to be measured for all stabilized nanobodies and compared to the measurement before stabilization.

We believe that taking affinity measurements for all wild type and stabilized nanobodies is beyond the scope of our study (most of the nanobodies in our list have never been characterized for affinity). Furthermore, part of our decision not to characterize affinity differences between wild type and stabilized nanobodies is that such measurements may not accurately represent binding potentials in the intracellular environment. In order to measure affinity, one would need to express and purify nanobodies. To do so, disulfide bonds must be maintained. Affinity measurements, in such an oxidized context, may not tell us what we would like to know as the stabilized nanobodies in an intracellular environment are not in an oxidized environment and do not have disulfide bonds. We believe our proposed stabilization method for intrabodies represents a simple and low investment means of potentially creating useful tools from otherwise poorly expressed reagents. However, utility will have to be assessed on a case-by-case basis.

4. At least one of the stabilize nanobodies needs to be tested against its endogenous target to determine if the nanobody binds sufficiently to immunoprecipitate or colocalize with target.

We have included an experiment which illustrates the ability of a mutationally stabilized nanobody to colocalize with an endogenous target, GFAP, in vivo in the retina (see Figure 6).

References:

1. Nguyen, Viet Khong, Raymond Hamers, Lode Wyns, and Serge Muyldermans. (2000). Camel heavy-chain antibodies: diverse germline VHH and specific mechanisms enlarge the antigen-binding repertoire. The EMBO Journal 19: 921– 930. https://doi.org/10.1093/emboj/19.5.921.

2. McMahon, Conor, Alexander S. Baier, Roberta Pascolutti, Marcin Wegrecki, Sanduo Zheng, Janice X. Ong, Sarah C. Erlandson, et al. (2018). Yeast surface display platform for rapid discovery of conformationally selective nanobodies. Nature Structural & Molecular Biology 25: 289–296. https://doi.org/10.1038/s41594-018-0028-6.